# Analysis of the Gut Mycobiome in Adult Patients with Type 1 and Type 2 Diabetes Using Next-Generation Sequencing (NGS) with Increased Sensitivity—Pilot Study

**DOI:** 10.3390/nu13041066

**Published:** 2021-03-25

**Authors:** Dominika Salamon, Agnieszka Sroka-Oleksiak, Artur Gurgul, Zbigniew Arent, Magdalena Szopa, Małgorzata Bulanda, Maciej T. Małecki, Tomasz Gosiewski

**Affiliations:** 1Department of Molecular Medical Microbiology, Chair of Microbiology, Faculty of Medicine, Jagiellonian University Medical College, 18 Czysta Street, 31-121 Krakow, Poland; dominika.salamon@uj.edu.pl (D.S.); agnieszka.sroka@uj.edu.pl (A.S.-O.); 2Center for Experimental and Innovative Medicine, University of Agriculture, Krakow1c Rędzina Street, 30-248 Krakow, Poland; artur.gurgul@urk.edu.pl (A.G.); zbigniew.arent@urk.edu.pl (Z.A.); 3Department of Metabolic Diseases, Faculty of Medicine, Jagiellonian University Medical College, 2 Jakubowskiego Street, 30-688 Krakow, Poland; magdalena.szopa@uj.edu.pl (M.S.); maciej.malecki@uj.edu.pl (M.T.M.); 4Department of Metabolic Diseases, University Hospital, 2 Jakubowskiego Street, 30-688 Krakow, Poland; 5Department of Epidemiology of Infections, Chair of Microbiology, Faculty of Medicine, Jagiellonian University Medical College, 31-121 Krakow, Poland; malgorzata.bulanda@uj.edu.pl

**Keywords:** diabetes, gut mycobiome, next-generation sequencing (NGS)

## Abstract

The studies on microbiome in the human digestive tract indicate that fungi could also be one of the external factors affecting development of diabetes. The aim of this study was to evaluate the quantitative and qualitative mycobiome composition in the colon of the adults with type 1 (T1D), *n* = 26 and type 2 (T2D) diabetes, *n* = 24 compared to the control group, *n* = 26. The gut mycobiome was characterized in the stool samples using the analysis of the whole internal transcribed spacer (ITS) region of the fungal rDNA gene cluster by next-generation sequencing (NGS) with increased sensitivity. At the L2 (phylum) level, Basidiomycota fungi were predominant in all 3 study groups. Group T1D presented significantly lower number of Ascomycota compared to the T2D group, and at the L6 (genus) level, the T1D group presented significantly lower number of *Saccharomyces* genus compared to control and T2D groups. In the T1D group, a significant positive correlation between total cholesterol and low-density lipoprotein cholesterol (LDL-C) levels and fungi of the genus *Saccharomyces,* and in the T2D group, a negative correlation between the total cholesterol level and *Malassezia* genus was found. The obtained results seem to be a good foundation to extend the analysis of the relationship between individual genera and species of fungi and the parameters determining the metabolism of carbohydrates and lipids in the human body.

## 1. Introduction

At the end of 2006, the World Health Organization (WHO) classified diabetes as the epidemic of the 21st century. In 2019, the International Diabetes Federation (IDF) reported that there were 463 million people suffering from diabetes and estimated that, by 2045, this number will increase by 51% and reach as many as 700 million [1]. For years, research has been conducted to identify the factors that induce changes in the body which lead to hyperglycemia and then to the development of diabetes symptoms. However, this is not a conclusive explanation of the mechanisms underlying both types of the disease and raises numerous hypotheses [2,3,4,5]. In this context, the microbiome of the digestive tract may be one of the external factors affecting the development of the disease and its course, as is evidenced by the results of many years of research analyzing the relationship of microorganisms inhabiting the intestines of people with type 1 (T1D) or type 2 diabetes (T2D). Yet, the majority of these analyses concentrate on bacteria, which in fact constitute the highest percentage of microorganisms, especially in the large intestine. Studies on patients with T1D have shown reduced variety and decreased gut bacterial microbiota stability and a relationship between the gut microbiota and innate immune response in these patients [5]. Analysis of gut microbiota in T2D patients indicates the role of intestinal permeability, caused by a reduced number of intestinal bacteria producing short-chain fatty acids (SCFAs), in low-grade chronic inflammation [6,7,8].

More and more frequent studies on fungi in the digestive tract of mammals indicate, however, that although the mycobiome is only 0.1% of the intestinal microbiome composition, its role in maintaining the homeostasis of the body seems to be significant [9,10,11,12]. A few studies which, in the analysis of the microbiome, focused on investigating fungi, demonstrated the presence of fungi in the digestive tract in both healthy people, as evidenced by the Human Microbiome Project [13], as well as in patients with inflammatory bowel disease (IBD). The gut mycobiota composition seems to be influenced by diet, including consumption of carbohydrates, which correlate positively with fungi of the genus *Candida* [14]. Additionally, an effective biological treatment applied to children with Crohn’s disease (CD) resulted in reducing the number of fungi of the genus *Candida* in the colon of these patients, which may suggest their significance in the development and course of CD [15]. Analysis of the gut microbiome in patients with diabetes, also taking fungi into consideration, was also conducted, showing that fungi of the genera *Aspergillus* and *Candida* (opportunistic fungal pathogens) were overrepresented in newly diagnosed type 2 diabetic subjects [16]. In our previous research on the quantitative evaluation of the genus *Candida* in the feces of patients with type 1 and type 2 diabetes, we demonstrated greater amounts of these fungi in patients with diabetes compared to the control group, and their amount seems to be associated with serum lipids in T2D patients [17].

Most of the gut mycobiome analyses to date have been based on the culture method or polymerase chain reaction (PCR) specific for taxa, however, increasingly more often, the techniques used include a high-throughput method based on next-generation sequencing (NGS), which makes use of nucleotide sequences of marker genes and allows for a more reliable taxonomic and phylogenetic analysis of microorganisms. The most commonly used molecular marker to determine the species affiliation of a given fungus is the internal transcribed spacer (ITS) genomic region in the rRNA operon encoding the fungal ribosomal DNA, which consists of: ITS1, 5.8S, and ITS2 regions [18,19,20].

The aim of our study was to evaluate the quantitative and qualitative fungal microbiome composition in the colon of adults with type 1 and type 2 diabetes using the analysis of the whole ITS region by the NGS technique with increased sensitivity. We have also attempted to establish a relationship between the profile of the gut mycobiota of patients and their clinical data.

## 2. Materials and Methods

### 2.1. Study Population

Seventy-six adults, aged 20 to 65 years, were included in the study. There were 50 diabetic patients, 26 with T1D and 24 with T2D, hospitalized with decompensated diabetes in 2012–2015 at the Department of Metabolic Diseases, University Hospital, Krakow, Poland. Inclusion criteria for the study were: age between 20 and 65 years, disease duration for at least 2 years, and: (1) for patients from the T1D group: initiation of insulin therapy in the first year after diagnosis, and (2) for patients from the T2D group: use of oral medications for at least 2 years after the diagnosis of diabetes. The exclusion criteria were: congenital and acquired immune deficiencies, any type of diabetes, latent autoimmune diabetes of adults (LADA), maturity-onset diabetes of the young (MODY), confirmed gastrointestinal infections, chronic inflammatory bowel disease (Crohn’s disease, ulcerative colitis), celiac disease, active cancer (especially gastrointestinal), renal failure, cirrhosis, pregnancy, antibiotic and antimycotic therapy within 30 days before drawing fecal samples (to exclude their influence on the composition of the microbiome), use of probiotics or prebiotic therapy within 30 days before drawing fecal samples, and lack of consent to participate in the study or withdrawal of consent during the study. The control group consisted of 26 healthy volunteers without antibiotic, antimycotic, probiotics, or prebiotic therapy within 30 days before drawing fecal samples.

The study was performed according to the Declaration of Helsinki and was approved by the Bioethical Committee of Jagiellonian University (No. KBET/81/B/2010). All the participants gave their written informed consent.

### 2.2. Materials

Stool samples from the individuals qualified for the study were analyzed. These samples were collected from patients and volunteers at the Department of Metabolic Diseases, University Hospital, Kraków, Poland, into appropriate 30 mL polypropylene containers (FL Medical, Padova, Italy) and immediately frozen at −80 °C, then delivered in deep-freeze conditions to the Department of Microbiology of the Jagiellonian University Medical College (JUMC), Kraków, Poland. At the same time, routine laboratory tests evaluating biochemical parameters were conducted for all patients: the assessment of glycated hemoglobin A1c (HbA1c), lipid profile (total cholesterol, high-density lipoprotein cholesterol (HDL-C), low-density lipoprotein cholesterol (LDL-C), and triglyceride levels (TG)), alanine aminotransferase (ALT), and creatinine levels, as well as estimated glomerular filtration rate (eGFR) calculated according to the Modification of Diet in Renal Disease Study Group formula. Age, body mass index (BMI), and disease duration were also recorded.

### 2.3. Library Preparation

At the Department of Microbiology JUMC, Kraków, Poland, fungal DNA was isolated from 76 stool samples using the Genomic Mini AX Stool Spin kit (A&A Biotechnology, Gdańsk, Poland) with the application of a preliminary procedure, as described by us earlier [15,21].

The next step involved the creation of an amplicon library of the whole ITS regions of the fungal rDNA gene cluster for each sample tested (Figure 1).

Since, as stated above, the mycobiome constitutes only 0.1% of the whole gut microbiome [9,11,12], some of the fecal samples did not demonstrate the presence of fungal genetic material. In order to solve this problem, we employed the nested method (according to the developed protocol—Table 1) to prepare genetic libraries of ITS in order to increase the sensitivity and specificity of the amplified fragments. To exclude the possibility of obtaining a PCR signal from potential contamination, a control was employed in the form of water, which was also subjected to amplification using the nested PCR technique.

The Illumina overhang adapter sequences: F: TCGTCGGCAGCGTCAGATGTGTATAAGAGACAG and R: GTCTCGTGGGCTCGGAGATGTGTATAAGAGACAG, were added to the 5′ end of the internal primers. Subsequently, the protocol for the MiSeq high-throughput sequencer (Illumina, San Diego, CA, USA) was followed [22].

### 2.4. Next-Generation Sequencing

Sequencing was carried out using the NGS method—sequencing by synthesis (SBS). The 10 pM library containing 76 pooled indexed samples with 10% spike-in PhiX control DNA was loaded onto the MiSeq (Illumina) apparatus. Sequencing was performed using the MiSeq Reagent Kit v3 (600 cycles). The sequencing procedure was performed in the Center for Medical Genomics OMICRON, Jagiellonian University Medical College, Krakow, Poland.

### 2.5. Bioinformatic and Statistical Analysis

The obtained raw sequencing reads were controlled for quality using FastQC software (Babraham Bioinformatics, Cambridge, UK). The reads’ quality was high, and no overrepresented primer or adapter sequences were detected. The sequences were subsequently analyzed using BaseSpace (Illumina) ITS Metagenomics software, which is a high-performance version of the Ribosomal Database Project (RDP) Classifier developed by Wang et al. [23]. The analysis with RDP included: screening the reads against primer sequences, discarding non-target reads, filtering the reads by base-call quality, length, and ambiguity, and merging the paired-end reads. Finally, the UCHIME (https://www.drive5.com/usearch/manual/uchime_algo.html, accessed on 23 March 2021) [24] software was used to detect chimeric reads. Processed reds were further assigned to the taxonomic classes using the RDP algorithm which is based on the Bayesian approach [23]. The RDP classification was made with respect to the UNITE database v7.2 (https://unite.ut.ee/, accessed on 23 March 2021) of the project described by Kõljalg et al. [25]. This version of the database contains 30,696 sequences, 59% of which are classified down to the species level. Counts of reads classified to the specific operational taxonomic units (OTUs) were used to asses Alfa and Beta diversity using MicrobiomeAnalyst software (https://www.microbiomeanalyst.ca/, accessed on 23 March 2021) [26]. Alpha diversity, expressed in the Shannon and Chao1 indexes, as well as observed diversity were assessed for significance using analysis of variance (ANOVA). The beta diversity evaluated using the Shannon index was subsequently tested for significance using permutational analysis of variance (PERMANOVA) and visualized with principal coordinates analysis (PCoA). To further analyze differences in amounts of separate fungi, DESeq2 software (https://bioconductor.org/packages/release/bioc/html/DESeq2.html, accessed on 23 March 2021) [27] was used to normalize read counts and to perform differential analysis. The obtained *p*-values were corrected for multiple testing using the false discovery rate (FDR) procedure [28]. Principal component analysis (PCA) and its visualization were also generated with DeSeq2 software based on counts of reads classified to separate fungi on the genus level.

Differences in clinical parameters (HDL, LDL, etc.) among groups were evaluated using ANOVA analysis. Further, the parameters were tested for distribution using the Shapiro–Wilk test and those deviating from normal distribution were analyzed using the Kruskal–Wallis test with post hoc analysis (for analysis of variability between the 3 study groups) and the Mann–Whitney test (for analysis of variability between the T1D and T2D groups). The dependence between amounts of specific fungi and clinical traits was evaluated using correlation analysis, in which traits deviating from normal distribution were analyzed using Spearman’s rank correlation coefficient. A *p*-value of less than 0.05 was assumed as significant. The chance of finding one or more significant differences in 33 performed statistical tests for clinical parameters was evaluated at 0.816 (81.6%). To account for this, a Sidak’s correction for multiple testing was applied which lowered the marginal *p*-value to 0.00155. In case of correlation analysis, we established a power of this analysis using G*Power v3.1.9.4 software (Heinrich Heine University, Düsseldorf, Germany) [29]. The power was estimated at 0.766 and 0.728 when analyzing a single group (*n* = 26 and *n* = 24, respectively), with alpha of 0.05 and detecting strong correlations (0.5).

## 3. Results

### 3.1. Characteristics of the Study Population

Clinical data of the studied groups of patients and control group are presented in Table 2. Statistically significant differences between the groups were found in terms of age, BMI, HDL-C, ALT, and the duration of the disease.

### 3.2. Metagenomic Sequencing

Sequencing of 76 fecal samples gave 10,106,525 reads, 132,813 reads on average, per sample (± 66,985). The cut-off was 20,000 reads per sample. As the maximum number of reads per sample was 320,031 and the minimum was 24,234, no sample was excluded from the final analysis. The median amounted to 126,716.

The obtained DNA sequences corresponded to 233 OTUs in total at the species level (L7). At the genus level (L6), 183 OTUs were detected. Details for the percentage of reads that mapped (classified) to specific OTUs at different taxonomic levels are presented in Table 3.

The reads were clustered during preliminary filtering for 97% identity, which is equivalent to sequence similarity at a given taxonomic level; therefore, due to the possibility of error in the interpretation of the obtained data, we did not analyze the mycobiota composition at the species level (L7), because the percentage of classified reads was below 97% (Table 3). Correlation analyses were conducted for selected clinical data and the numbers of only those OTUs which were clearly assigned to a specific species.

Alpha diversity expressed in Chao1 and Shannon indices was comparable within groups T1D and T2D, and higher than the control group, and these differences were not statistically significant (*p* > 0.05); in observed OTUs, it was higher in group T1D, but the difference was just at the limit of statistical significance (*p* = 0.47) (Figure 2, Figure 3 and Figure 4).

Beta diversity was similar in the control group and T2D (*p* > 0.05). A significantly smaller distance between OTUs (a closer phylogenetic relationship of OTUs in the samples) was found in the T1D group (*p* < 0.011, PERMANOVA) (Figure 5).

We conducted a systematic assessment of the fungal profile at the taxonomic level (L2) and at L6 (at L7, below 95% of the reads were classified) in order to create a general picture and detailed analysis of differences in the gut microbiota composition of the samples tested.

At L2 (phylum), we identified OTUs corresponding to 15 types, but only 4 types reached the percentage of above 1% (Basidiomycota, Ascomycota, Chlorophyta, and an unidentified phylum). Chlorophyta is a taxon of green algae belonging to eukaryotic organisms, selected species of which can form symbiotic forms with fungi, known as lichens [30]. Due to the fact that this phylum was identified in the course of sequencing, probably due to the similarity of the selected DNA fragment to fungal sequences, it will not be analyzed later in this study.

In all 3 groups under study, Basidiomycota fungi were predominant, and they accounted for the majority of the mycobiota of the samples tested (Figure 6).

When comparing the numbers of the above-mentioned mycobial types in the 3 analyzed groups, we found a statistically significant difference as regards Ascomycota between T1D and T2D groups (adjusted *p*-value; *p* adj 0.033).

Due to the large number of OTUs, only statistically significant data at the L6 level are presented and all taxa with a relative percentage below 1% are shown as “other”. At this level (L6—genus), OTUs corresponding to 183 types of fungi were identified, as shown in Figure 7. All the three groups analyzed were dominated by fungi of the genus Malassezia, which accounted for the majority of the mycobiota in the samples studied, and their relative percentage did not differ significantly in any of the studied groups. On the other hand, significant differences were demonstrated as regards the amounts of other genera, as shown in Table 4.

The relative percentage of the genus *Rhodotorula* demonstrated quantitative differences between the control and T1D as well as T1D and T2D groups, but their significance was suggested only by the point probability value (*p*-value, respectively: 0.024 and 0.034); however, after correction using the FDR procedure, these results turned out to be statistically insignificant (*p* > 0.05).

The proportions of the following three types of fungi did not exceed 1% in any of the study groups, but because of their clinical relevance, we evaluated the differences regarding their relative percentages in the analyzed groups. The genus *Aspergillus* was most predominant in control group (0.82%), and least common in the T1D group (0.006%), and this difference is statistically significant in relation to the control group (*p* adj 0.008), while in the T2D group, the relative percentage amounted to 0.013%. The genus *Cryptococcus* amounted to: in the control group, 0.19%, T1D group, 0.43%, and T2D group, 0.01%. We demonstrated significant differences in the amounts of this genus between the groups: control and T1D (*p* adj < 0.0001) as well as T1D and T2D (*p* adj < 0.0001). The relative percentage of fungi of the genus *Candida* was 0.12% in the control group, while in T1D it was 0.28%, and in T2D, 0.08%, and it was significantly different between them (*p* adj 0.04).

### 3.3. Correlation Analysis

A comparison of the relative percentage of the fungal genera and data, such as: age, BMI, disease duration, and selected clinical parameters, demonstrated significant dependencies within both groups of patients with diabetes. In the T1D group, a significant negative correlation was found between the HDL-C level and fungi of the genus *Cladosporium* (*r_s_* = −0.43, *p* = 0.03) and a positive correlation between total cholesterol and LDL-C levels and fungi of the genus *Saccharomyces* (respectively: *r_s_* = 0.39, *p* = 0.04, and *r_s_* = 0.49, *p* = 0.01), as well as ALT versus *Cryptococcus* (*r_s_* = 0.49, *p* = 0.01). In the T2D group, a positive correlation was observed between the HDL-C level and fungi of the genus *Rhodotorula* (*r_s_* = 0.44, *p* = 0.03) and a negative correlation between BMI and *Penicillium* (*r_s_* = −0.45, *p* = 0.025) and between the total cholesterol level and *Malassezia* (*r_s_* = −0.48, *p* = 0.017).

Analysis of correlations concerning the genus *Malassezia* showed several interesting relationships at the L7 level. In the T1D group, the species *M. globosa* correlates positively with HbA1c (*r_s_* = 0.47, *p* = 0.01), and the species *M. yamatoensis* correlates negatively with the LDL-C level (*r_s_* = −0.412, *p* = 0.036). Whereas, in the T2D group, negative correlations were found between *M. restricta* and the LDL-C level (*r_s_* = −0.51, *p* = 0.011), total cholesterol level (*r_s_* = −0.56, *p* = 0.004), and TG (*r_s_* = −0.406, *p* = 0.049). However, between *M. sympodialis* and the HDL-C level, there was a positive correlation (*r_s_* = 0.43, *p* = 0.036). No significant dependencies were demonstrated at the L7 level in relation to the species of fungi that could be identified, from the genera *Saccharomyces, Cryptococcus,* and *Rhodotorula.*

## 4. Discussion

There are still few studies which analyze the gut mycobiome in patients with diabetes. Observations in such patients most often concerned fungi of the genera *Candida* or *Aspergillus* and indicated their outgrowth in the gastrointestinal tract in patients with poor glycemic control [16,17,31,32]. Additionally, studies assessing the whole gut mycobiome mainly concentrate on patients with type 2 diabetes and possible differences in the mycobiome composition and its interactions with bacteriobiome in the complications resulting from this type of diabetes [16,33,34]. In our study, we employed the NGS technique, allowing a detection of non-culturable fungi in the human intestine, which were previously unknown and, also, we evaluated the profile of the gut mycobiota of adult patients with both type 1 and type 2 diabetes. The results presented by us are the first such comprehensive analysis concerning the gut mycobiome in patients with two types of diabetes. Furthermore, due to the fact that the mycobiome is only 0.1% of the whole microbiome [9,11,12], which is reflected in the lack of an ITS sequence amplification signal in some of the samples tested, we had decided to employ the Nested PCR technique, in order to increase the sensitivity of fungal detection in the samples under study, which is also the first time this research approach has been applied for the analysis of the microbiome in patients with diabetes.

The group that differs significantly from the others in terms of age, BMI, and HDL-C levels is the T2D group (Table 2). Hence, significant differences are to be expected as regards the gut mycobiome composition in these patients, especially as there are studies suggesting a relationship between the gut mycobiota and age [35] or obesity [36]. However, contrary to these reports and in opposition to the results obtained by us in the course of analysis of the gut bacteriobiome in adult patients with diabetes [7], in the present study, we demonstrated differences in the composition of the mycobiome and biodiversity in relation to the other groups primarily in the T1D group, and not T2D. Therefore, it should be assumed that the gut mycobiome profile is also significantly affected by other factors.

Biodiversity analysis did not show statistically significant differences concerning the alpha diversity of the studied samples. Similar results were obtained by Al Bataineh et al. among Emirati subjects with type 2 diabetes compared to a control group [34]. Contrary to our observations, the results by Jayasudha et al., assessing the gut mycobiome in Indian patients with type 2 diabetes, show a statistically significant difference in the alpha diversity of samples from patients with type 2 diabetes, both without and with diabetic retinopathy compared to healthy individuals [33].

On the other hand, a comparison of beta diversity in the 3 groups indicates a significantly greater phylogenetic affinity of microorganisms between individual fecal samples of people from the T1D group, which is therefore more homogeneous and significantly different from the T2D and control groups. Perhaps this has to do with the treatment method and specific diet (reduction in carbohydrates and fats) and a significantly longer disease duration (from adolescence), which results in greater discipline as regards glycemic control (Table 2). One of the limitations of our study is the lack of detailed data on the diet and medications taken by the patients. However, taking into account the cultural and geographical uniformity of the three groups analyzed, as well as the fact that, in patients with type 1 diabetes, treatment with insulin and appropriate diet are introduced when the disease is diagnosed and the patients usually follow the regimen, it can be assumed that these two factors contribute to the substantial homogeneity of the T1D group in terms of biodiversity and cause a visible difference in beta diversity of this group compared to the two other groups.

Many studies assessing the gut mycobiota demonstrate a high proportion of Ascomycota fungi and, to a smaller degree, Basidiomycota fungi [13,16,36,37]. In our analysis, the significantly dominant phylum in all 3 treatment groups (T1D, T2D, and control group) was Basidiomycota. It could have been influenced by the method of preliminary preparation for the microbial DNA isolation itself developed by Gosiewski et al. [21], which makes use of aggressive destruction of the cell wall. In fungi belonging to the phylum Basidiomycota, the cell wall is relatively thick, and the above-mentioned preliminary procedure allows for its effective destruction and facilitates the release of DNA of these fungi. Nevertheless, similar results to ours were obtained by Jayasudha et al., who employed primers for the ITS 2 region in their study and also found that Basidiomycota was the most dominant phylum in the three examined groups (control, T2D without, and T2D with diabetic retinopathy) [33]. The possibility of detecting more DNA of the phylum Basidiomycota using the ITS 2 region was indicated by Hamad et al. [37], while the team of Toju, in their analyses of sets of several primers, suggested that the application of selected primers for the whole ITS region will increase the reliability of fungal species identification [20]. In our study, we used the ITS1-F sequence as a forward primer and the ITS4 sequence as a reverse primer for the whole ITS region (Figure 1), which enabled us to assume that, with a high degree of probability, the results obtained by us reflect the actual composition of the mycobiome.

Accordingly, at the L6 level, the dominant genus in all three study groups, without significant quantitative differences, was *Malassezia*. Species of this genus are mostly characteristic of the skin, but the development of research methods independent of culturing, which proves difficult in this case, allowed to show the presence of these fungi in other niches, including the human gastrointestinal tract [38].

Many significant differences concerning the numbers of fungi inhabiting the colon of the studied subjects involve saprophytic microorganisms, coming from the external environment. However, attention should be drawn to the much smaller, statistically significant compared to the other groups, number of fungi of the genus *Saccharomyces* in samples from patients with T1D (Figure 6). Perhaps, it is associated with strict compliance with the diet on the part of these patients. Hoffmann et al., in their study concerning the relationship between the composition of the gut archeobiome and gut mycobiome and diet, suppose that the greater number of fungi of this genus might be associated with high consumption of beer and bread [14]. It is interesting that the amounts of fungi of the genus *Ganoderma* is significantly low in the group of patients with T2D. This genus encompasses several dozen species, many of which are cosmopolitan and saprophytic, some cause diseases in trees, and others are linked to potential therapeutic effect due to their antioxidant and anti-inflammatory properties [39,40]. In type 2 diabetes, the phenomena of metabolic endotoxemia and metabolic bacteremia are described, which result from increased intestinal permeability to potentially pathogenic Gram-negative bacteria, which leads to low-grade inflammation and insulin resistance [41]. In our study of the bacteriobiome in adult patients with diabetes, we had also found a significantly lower proportion of bacteria of the genus *Roseburia* in the group of patients with T2D. These bacteria play a part in maintaining intestinal wall integrity by producing short-chain fatty acids (SCFAs) [7]. Perhaps such inflammation in patients with T2D may also be associated with reduced numbers of fungi of the genus *Genoderma* with anti-inflammatory and antibacterial potential [40], especially since there is a specific fungal-bacterial interaction, which is pointed out by, among others, Chin et al., who cite examples of studies proving that *Saccharomyces boulardii* secretes enzymes involved in the deactivation of the toxins produced by *Clostridioides difficile* and *Escherichia coli* [11].

The low relative percentages of the known genera of fungi: *Candida*, *Cryptococcus, Penicillium*, or *Aspergillus* (generally reported in the majority of other studies), that were demonstrated in all 3 groups in this study, should be attributed to the use of the NGS technique, which enables the identification of numerous non-culturable fungi, which probably reflects the actual proportion of the individual genera of fungi present in the gut mycobiota. Interestingly, in contrast to our earlier pilot study by Gosiewski et al. [17], the present analysis showed the lowest number of fungi of the genus *Candida* in the T2D group (and not in the control group), and furthermore, we did not confirm a significant correlation between this type of fungus and lipid levels. This can be attributed to the fact that the previous study [17] was based on absolute numbers of *Candida* cells expressed in terms of colony-forming units per gram (CFU/g), and in NGS, percentage results were analyzed, i.e., relative to the entire pool of reads obtained. Moreover, in this study, the treatment groups were more numerous: T2D (24 vs. 17) and control (26 vs. 17).

The small size of the studied groups is another limitation of our research. With such a number of samples tested, significant differences in the values of variables such as age or BMI in the control, T1D and T2D groups can be confounding factors of the correlation analysis. Nevertheless, we attempted to evaluate the correlation demonstrated between the number of specified genera of fungi and the clinical parameters because the calculated power of the correlation analysis in a single group was estimated at >0.7, which can be considered satisfying for the detection of strong correlations. However, it is necessary to confirm our observations presented below on a larger number of samples.

When analyzing the relationships between age, BMI, and selected clinical parameters, we discovered a significant positive correlation between fungi of the genus *Saccharomyces* and total cholesterol and the LDL-C level in the group of patients with T1D. However, in the study in which streptozotocin-diabetic mice (model of T1D) were treated with *Saccharomyces boulardii* THT 500,101 strain, Albuquerque et al. observed amelioration of dyslipidemia, but these results concerned only the reduction in the TG level [42]. The contradiction in both observations may also stem from the fact that, in our study, the positive correlation concerns the whole genus *Saccharomyces*, and not a particular strain which has an additional probiotic effect. However, the negative correlation found by us for fungi of the genus *Penicillium* and the BMI value in the group T2D confirms the results by Rodriguez et al., who observed that the relative abundance of the genus *Penicillium* correlated negatively with parameters of body fatness such as BMI, fat mass, android fat mass, and hip circumference [36].

When studying the gut mycobiome in patients with diabetes, we revealed interesting dependencies involving lipid profile and the amounts of selected species of fungi of the genus *Malassezia*. In both T1D and T2D, we observed a negative correlation between fungi of the genus *Malassezia* and selected fractions of lipid profile, apart from *M. sympodialis* in patients with T2D, which correlates positively with HDL-C. The genus *Malassezia* comprises species of lipid-dependent fungi, which obtain useful lipids from the environment using lipolytic enzymes such as lipase, esterase, phospholipase, and lysophospholipase, which may act as virulence factors. Therefore, these fungi are considered to be the cause of many inflammatory skin diseases, including seborrheic dermatitis, due to the presence of sebaceous glands, which are the source of lipids that these fungi require to build the outer layer of the cell wall [43]. A significant proportion of fungi of the genus *Malassezia* in the colon of people from all groups that were studied by us and a negative correlation between their number and the level of selected fractions of serum lipid profile may point to favorable conditions for the development of these fungi in the human gastrointestinal tract. This is also confirmed by the study by Spatz and Richard, in which the authors point to food and other microorganisms as the source of lipids for *Malassezia* sp., which are easily acquired by these fungi thanks to the activity of bile salts [38]. This may also be a premise for the possible future use of the properties of fungi of the genus *Malassezia* to regulate lipid profile in people with dyslipidemia. Additionally, in the T1D group, we determined a positive correlation between HbA1c and *M. globosa*. Aykut et al. have recently described their research concerning the gut mycobiome in individuals with pancreatic ductal adenocarcinoma (PDAC), in which they prove that fungi migrate from the gut lumen to the pancreas. They discovered a marked increase in the intra-tumoral *M. globosa* in human PDAC and in mouse models [44]. Presumably, this fungal species can not only directly damage pancreatic cells, causing them to malfunction, but also affect other cells of the human body [38]. Therefore, it can be assumed that it is capable of triggering an abnormal reaction of cells to insulin supplied from the outside and cause disorders in glucose metabolism.

## 5. Conclusions

Although fungi comprise a small percentage of the gastrointestinal tract microorganisms, their role seems to be significant. The results of our analysis prove that there are differences in the profile of the gut mycobiota of diabetic patients compared to the control group. Further research is needed, at lower taxonomic levels (L7) and on a greater number of samples, especially coming from patients with type 2 diabetes. The results obtained by us seem to be a good foundation to extend the analysis of the relationship between individual genera and species of fungi and the parameters determining the metabolism of carbohydrates and lipids in the human body. It also seems important to study fungal–bacterial interactions, as the data obtained may allow to develop individual therapies modifying both bacteriobiota and mycobiota of the human intestine and, hence, facilitating glycemic control and lipid profile in diabetic patients.

## Figures and Tables

**Figure 1 nutrients-13-01066-f001:**
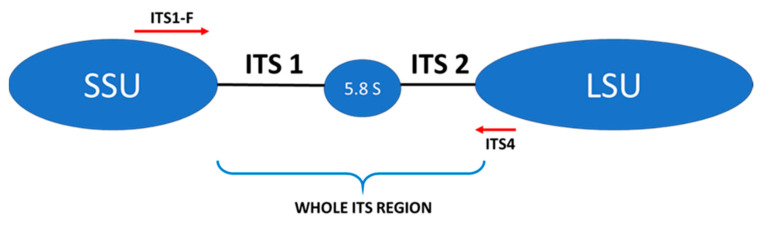
Schematic structure of the internal transcribed spacer (ITS) region and location of internal primers used in this study: ITS1-F, ITS 4—primers, LSU—large ribosomal subunit, SSU—small ribosomal subunit.

**Figure 2 nutrients-13-01066-f002:**
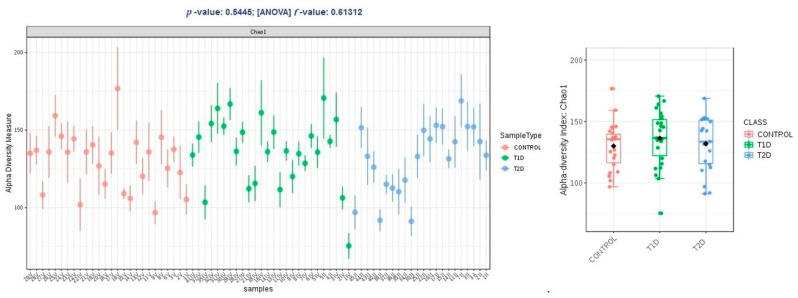
Alpha diversity expressed in Chao1 index.

**Figure 3 nutrients-13-01066-f003:**
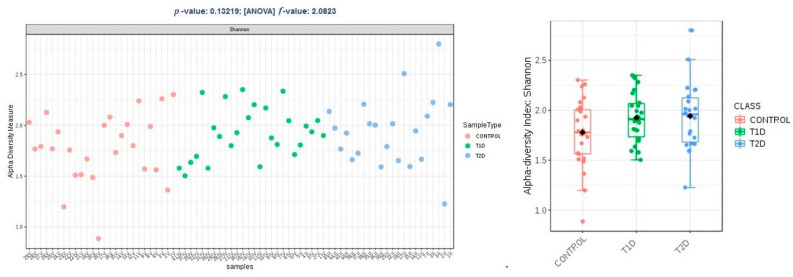
Alpha diversity expressed in Shannon index.

**Figure 4 nutrients-13-01066-f004:**
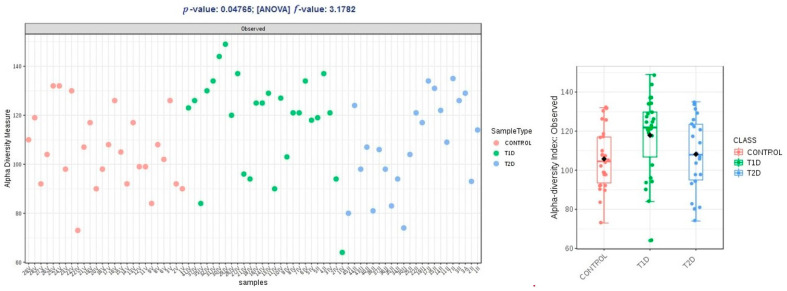
Alpha diversity expressed in observed OTUs (operational taxonomic units).

**Figure 5 nutrients-13-01066-f005:**
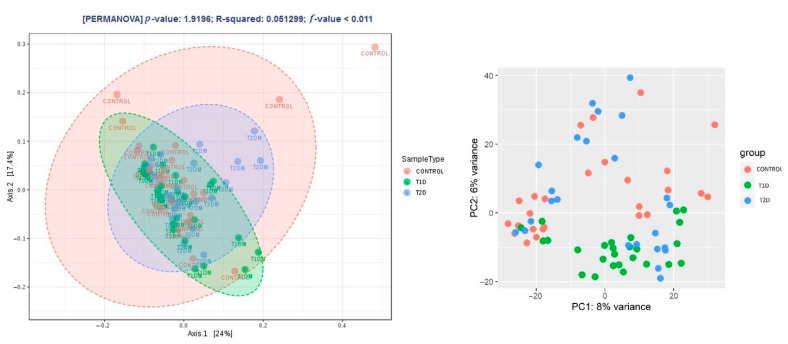
Beta diversity expressed with the Jensen–Shannon distance and the principal coordinates analysis (PCoA) and the principal component analysis (PCA) method of the DeSeq2 software.

**Figure 6 nutrients-13-01066-f006:**
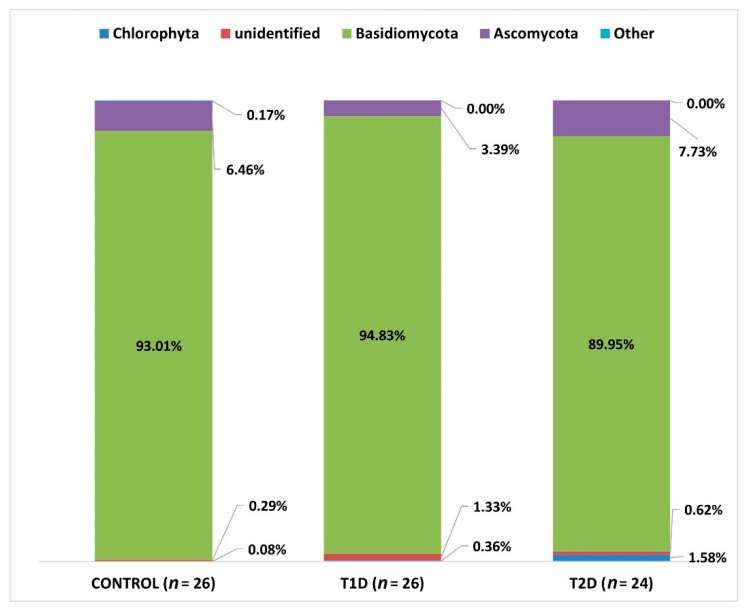
Fungal profiles in the diabetes and control groups at the phylum level (L2).

**Figure 7 nutrients-13-01066-f007:**
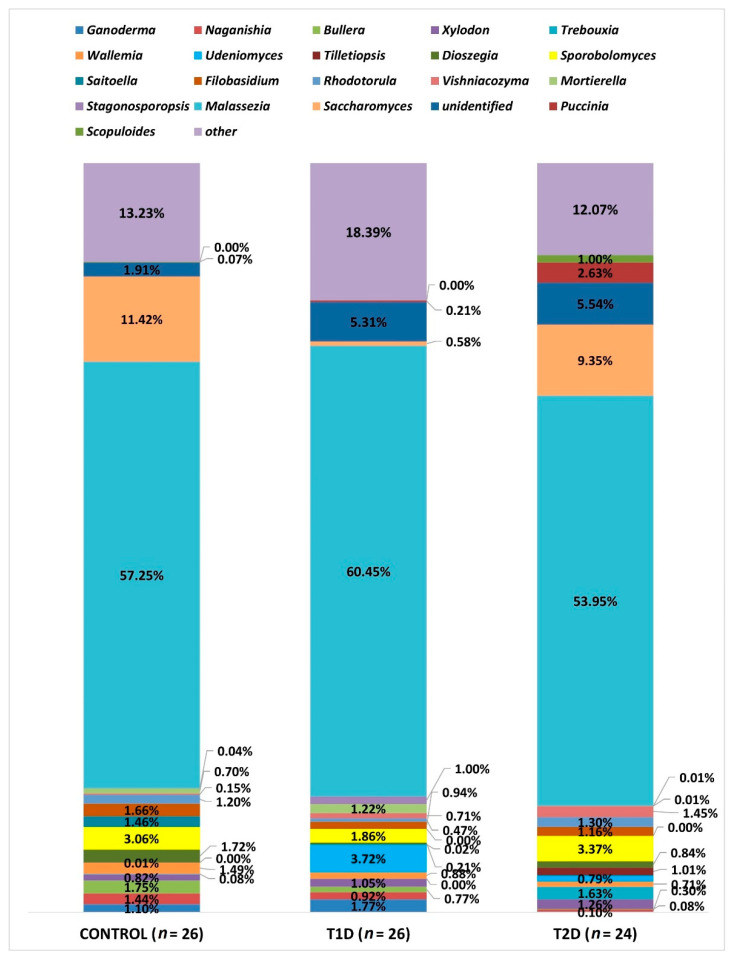
Fungal profiles in the diabetes and control groups at the genus level (L6).

**Table 1 nutrients-13-01066-t001:** Primer sequences (Genomed, Warsaw, Poland), reaction mixtures, and thermal amplification programs used in the study.

Primer Sequence 5′→3′	Reaction Mixture		Thermal Amplification Program
External primers ^a^ F: AAATGCGATAAGTAATGTGAATTGCAGAATT R: TTACTAGGGGAATCCTTGTTAGTTTCT	Water Kapa ^c^ Primer F (10 µM) Primer R (10 µM) DNA	2.0 μL 5.0 μL 0.5 μL 0.5 μL 2.0 μL	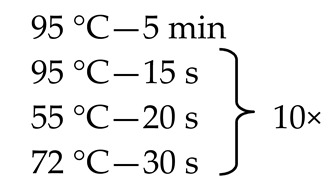
Internal primers ^b^ ITS1-F(F): CTGGTCATTTAGAAGTAA ITS4 (R): TCCTCCGCTTATTGTATGC	Water Kapa ^c^ Primer F (10 µM) Primer R (10 µM) DNA	9.5 μL 12.5 μL 0.5 μL 0.5 μL 2.0 μL	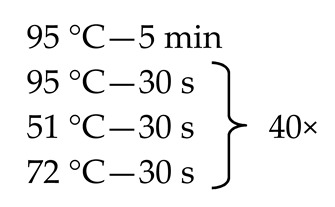

^a^ Own study; ^b^ Reference [20]; ^c^ 2 × KAPA HiFi HotStart ReadyMix (Roche Kapa Biosystems, Wilmington, NC, USA).

**Table 2 nutrients-13-01066-t002:** Clinical characteristics of the study groups.

Parameters	CONTROL (*n* = 26)	T1D (*n* = 26)	T2D (*n* = 24)	*p*-Value
F:M	19:7	20:6	9:15	-
Age, years	36 (31−46.5)	33(30−47)	56 (56.25−62.75)	<0.001 ^a^
BMI, kg/m^2^	23.1 (22.2−24.6)	22.2 (20.3−25)	27.2 (25−28.7)	<0.001 ^a^
HbA_1c_, %	5.35(5.2−5.5)	7.95(6.77−9.65)	7.1(6.41−8.56)	<0.001 ^b^
Total cholesterol, mmol/L	5.2(4.92−5.75)	5.0 (4.12−5.42)	4.82 (4.04−5.9)	0.458
HDL-C, mmol/L	1.8 (1.5−1.9)	1.6 (1.42−2.0)	1.08 (0.87−1.2)	<0.001 ^a^
LDL-C, mmol/L	3.15 (2.72−3.55)	2.7 (2.3−3.25)	2.94 (2.49−3.77)	0.215
TGs mmol/L	0.8 (0.69−1.19)	0.8 (0.65−1.35)	1.72 (1.4−2.29)	0.274
ALT, U/L	17 (13.2−19.85)	14 (11.2−19.5)	24.5 (20.5−35)	<0.001 ^a^
Creatinine, μmol/L	60 (56−66)	58 (55−68)	59 (56−65)	<0.39
eGFR (MDRD), mL/min/1.73 m^2^	115.3 (118.6−110.8)	118.7 (121.3−111.25)	108.2 (110.5−103.9)	0.06
Duration of diabetes, years	-	15.5 (5.5−22.75)	5.5 (2.25−10)	0.004 ^c,^*

Data are presented as median (interquartile range). A *p*-value of less than 0.05 is considered statistically significant. ^a^ Difference between group type 2 diabetes (T2D) and groups: T1D and control (Kruskal–Wallis test with post hoc analysis). ^b^ Difference between control group and groups: T1D and T2D (Kruskal–Wallis test with post hoc analysis). ^c^ Difference between group T1D and group T2D (Mann–Whitney test), * Difference in the duration of diabetes is not significant after Sidak’s correction for multiple testing. Abbreviations: ALT, alanine aminotransferase; BMI, body mass index; eGFR; estimated glomerular filtration rate; F, female; HbA1c, hemoglobin A1c; HDL-C, high-density lipoprotein cholesterol; LDL-C, low-density lipoprotein cholesterol; M, male; MDRD, Modification of Diet in Renal Disease Study Group; T1D, type 1 diabetes; T2D, type 2 diabetes; TGs, triglycerides.

**Table 3 nutrients-13-01066-t003:** Phylogenetic summary of the obtained reads.

Taxonomic Level	CONTROL		T1D		T2D	
	No. of Classified Reads	Percent of Reads	No. of Reads	Percent of Classified Reads	No. of Reads	Percent of Classified Reads
Kingdom	2,896,459	96.92	4,142,146	96.56	2,782,278	95.83
Phylum	2,880,932	99.48	4,118,805	99.35	2,765,589	99.33
Class	2,862,193	99.33	4,079,130	99.04	2,732,493	98.66
Order	2,813,641	98.23	4,021,210	98.69	2,700,384	98.75
Family	2,784,972	99.07	3,976,863	99.01	2,670,820	98.80
Genus	2,751,978	98.91	3,887,741	98.14	2,652,556	99.31
Species	2,492,288	91.98	3,684,955	94.59	2,381,592	90.98

**Table 4 nutrients-13-01066-t004:** Significant differences between study groups as regards the amounts of the genera with a relative percentage about 1% in at least one study group.

Genus	Difference Between Groups (Relative Percentage)	Adjusted *p*-Value
*Saccharomyces*	Control (11.42%) vs. T1D (0.58%) T1D (0.58%) vs. T2D (9.35%)	<0.001 <0.0001
*Dioszegia*	Control (1.72%) vs. T1D (0.21%)	0.005
*Xylodon*	Control (0.81%) vs. T1D (1.05%)	0.005
*Mortierella*	Control (0.7%) vs. T1D (1.22%)	0.008
*Naganishia*	Control (1.44%) vs. T2D (0.3%) T1D (0.92%) vs. T2D (0.3%)	<0.0001 <0.0001
*Udeniomyces*	Control (0.01%) vs. T1D(3.72%) Control (0.01%) vs. T2D (0.79%) T1D (3.72%) vs. T2D (0.79%)	<0.0001 <0.0001 0.007
*Bullera*	Control (1.75%) vs. T2D (0.08%) T1D (0.77%) vs. T2D (0.08%)	<0.0001 <0.001
*Tilletiopsis*	Control (0.002%) vs. T2D (1.01%) T1D (0.02%) vs. T2D (1.01%)	<0.0001 <0.0001
*Saitoella*	Control (1.46%) vs. T2D (0.002%)	<0.001
*Ganoderma*	Control (1.1%) vs. T2D (0.1%) T1D (1.77%) vs. T2D (0.1%)	0.013 0.02
*Vishniacozyma*	Control (0.15%) vs. T2D (1.45%)	0.04
*Wallemia*	Control (1.49%) vs. T2D (0.71%)	0.04

## Data Availability

The data presented in this study are not publicly available due to the fact of confidentiality reasons. These data are available on request from the corresponding author.

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
