# Peer review of "Analysis of the Gut Mycobiome in Adult Patients with Type 1 and Type 2 Diabetes Using Next-Generation Sequencing (NGS) with Increased Sensitivity—Pilot Study"

_nutrients, 2021, doi:10.3390/nu13041066_

Round 1

Reviewer 1 Report

My major concern is the study population in this study. Several key variables such as Age , gender, BMI were significantly different among control, T1D and T2D groups. Those variables were shown significant impact on the gut mycobiome. Therefore, with such a small sample size, I doubt that any findings in this article were able to adjust for those confounding factors.   

Author Response

Response to Reviewer 1 Comments

Dear Reviewer,

We thank you for your valuable comments to our manuscript.

We  carefully  considered  your  comments  and  revised  the  paper  according  your suggestions. We responded to your comments below.

Point 1: My major concern is the study population in this study. Several key variables such as Age , gender, BMI were significantly different among control, T1D and T2D groups. Those variables were shown significant impact on the gut mycobiome. Therefore, with such a small sample size, I doubt that any findings in this article were able to adjust for those confounding factors.  

Response 1: We agree with the Reviewer. The small size of the study groups is a limitations of our research. It was very difficult to recruit patients to the groups which were studied in our project because candidates often did not meet the inclusion criteria to eliminate confounding factors and could not be enrolled in a research. Many of the similar studies that we refer to in our article were realized on a similar number of patients (i.e. 12-48 samples), e.g. references no. 33, 34, 36.

Therefore, to account for sample size in our study and to evaluate critically our data we introduced correction for multiple testing in clinical parameters analysis and performed statistical power analysis for correlation. Most of differences found in clinical parameters were also significant after Sidak’s correction for multiple testing. We have added supplementary information to the Bioinformatic and Statistical Analysis section (lines 198-204) and in the Table 2. The calculated power of correlation analysis in single group was estimated at 0.766 and 0.728, which is satisfying when detecting strong correlations.

Also, we have edited our Discussion and added the following paragraph:

The small size of the studied groups is another limitation of our research. With such a number of samples tested, significant differences in the values of variables such as age or BMI in the control, T1D and T2D groups can be confounding factors of the correlation analysis. Nevertheless, we attempted to evaluate the correlation demonstrated between the number of specified genera of fungi and the clinical parameters because the calculated power of the correlation analysis in a single group was estimated at > 0.7, which can be considered satisfying for the detection of strong correlations. However, it is necessary to confirm our observations presented below on a larger number of samples.”(lines 461-468).

Yours sincerely,

Tomasz Gosiewski

Reviewer 2 Report

The authors conduct a pilot study and aimed to evaluate the quantitative and qualitative mycobiome composition in the colon of 26 adults with type 1, 24 with type 2 diabetes, and 26 with control group using the analysis of the whole ITS region by the next generation sequencing (NGS) with increased sensitivity.

Comments:

1.Line 231, but these differences were not statistically significant (p > 0.05), (Figures 2, 3, 4).

Figure 4, the p value is 0.04765; a p value of less than 0.05 was assumed as significant. ??

2.Line 313, (R= -0.43, p = 0.03)

Spearman's rank correlation coefficient often denoted by the Greek letter ρ (rho) or as rs.

3.line 270 and Figure 7

Maybe the statistically significant differences in the 3 analyzed groups could be presented or summarized in Table, as appropriate, to allow a rapid assessment.

Reviewer 3 Report

Line 87: Please start the sentence with the number in its text form. In this same line the following sentence starts without a capital letter.

Lines 193, 194 and 195 seems to be text from the paper layout instructions. I suppose it should be removed.

Table 2- It makes more sense if the comparison between groups that results in the p-value should be done between control group and the patient groups (T1D and T2D), than between T1D and T2D. At least, both comparisons should be made.

As limitations of the study should be added the low sample number.

Considerations about using NGS in this approach (gut mycobiome characterization) should also be added in discussion.

Round 2

Reviewer 1 Report

Thank the authors for their responses to my previous comments. The statistics have been much more improved. In addition, I suggest the authors to perform a multivariable regression using each taxa in table 4 as outcome. The predicting variables should include diabetes status, BMI, gender and age.  
